# Developing a patient flow visualization and prediction model using aggregated data for a healthcare network cluster in Southwest Ethiopia

**Balew Ayalew Kassie** [1]*, **Geletaw Sahle Tegenaw**[1,2]

**1** Faculty of Computing, Jimma Institute of Technology, Jimma University, Jimma, Ethiopia, **2** Artificial Intelligence & Biomedical Imaging Research Lab, Jimma Institute of Technology, Jimma University, Jimma, Ethiopia

* balew.ayalew@ju.edu.et

## Abstract

A health information system has been created to gather, aggregate, analyze, interpret, and utilize data collected from diverse sources. In Ethiopia, the most popular digital tools are the Electronic Community Health Information System and the District Health Information System. However, these systems lack capabilities like real-time interactive visualization and a data-driven engine for evidence-based insights. As a result, it was challenging to observe and continuously monitor the flow of patients. To address the gap, this study used aggregated data to visualize and predict patient flow in a South Western Ethiopia healthcare network cluster. The South-Western Ethiopian healthcare network cluster was where the patient flow datasets were collected. The collected dataset encompasses a span of 41 months, from 2019 to 2022, and has been obtained from 21 hospitals and health centers. Python Sankey diagrams were used to develop and build patient flow visualizations. Then, using the random forest and K-Nearest Neighbors (KNN) algorithms, we achieved an accuracy of 0.85 and 0.83 for the outpatient flow modeling and prediction, respectively. The imbalance in the data was further addressed using the NearMiss Algorithm, Synthetic Minority Oversampling Technique (SMOTE), and SMOTE-Tomek methods. In conclusion, we developed a patient flow visualization and prediction model as a first step toward an end-to-end effective real-time patient flow data-driven and analytical dashboard in Ethiopia, as well as a plugin for the already-existing digital health information system. Moreover, the need for and amount of data created by these digital tools will grow along with their use, demanding effective data-driven visualization and prediction to support evidence-based decision-making.

## Author summary

Predicting patient flow is essential for improving efficiency, resource allocation, and process optimization. However, the majority of existing literature targets exploring the length

**Data Availability Statement:** We would like to provide a Data Availability Statement for our manuscript. The data used in our research was obtained from 21 hospitals and health centers in a

South Western Ethiopia healthcare network cluster from 2019 to 2022, spanning 41 months. The data was collected in accordance with institutional review board regulations and confidentiality agreements to protect the privacy of the patients involved. We are committed to promoting data transparency and availability in scientific research. Researchers who are interested in obtaining the datasets can do so through the Capacity Building and Mentorship Partnership (CBMP) Program or by submitting a direct request to the Ministry of Health using the relevant application (https://www.moh.gov.et/site/projects-3-col/dhis2 or https://ndmc.ephi.gov.et/). Please also confirm that the authors did not have any special access or request privileges that others would not have.

**Funding:** The authors received no specific funding for this work.

**Competing interests:** The authors have declared that no competing interests exist.

of stay, waiting time, treatment time, test turnaround time, and boarding time within a single healthcare institution and between different departments within the institution (i.e., intra-patient flow analysis). Whereas little is explored about the flow of patients with the network structure or organization. Our study tried to investigate the flow of patients in the Healthcare Network Cluster in Southwest Ethiopia, where healthcare is structured in a 3-tier fashion and the referral patient flow starts from the lower primary healthcare level to the secondary level on the way to the tertiary level or a specialized hospital. The inter-patient flow analysis can help identify areas for improvement in the coordination of care and the efficient use of resources by tracking the flow of patients from one primary healthcare provider to specialized care and analyzing the characteristics of that flow. Furthermore, our study only takes into account direct flow patients who use or follow the traditional healthcare system structure, not indirect flows or alternative forms of healthcare access.

## 1. Introduction

Ethiopia's health system is structured into a three-tier structure, such as primary, secondary, and tertiary. Primary hospitals and primary health care units (Health Centers and Health Posts) are considered to be at the primary level of treatment, followed by secondary (General Hospitals) and tertiary (Tertiary Hospitals) [1,2].

The Ethiopian health system implemented the Health Information System (HIS), which incorporates data from population-based and institution-based data sources, to assist decision-making at all levels [3,4]. The health management information system or institution-based HIS is a system that collects, compiles, aggregate, analyze, interpret, and use data generated from various sources. It is used to manage health information at the community and facility level [5,6]. Among different digital tools, the Electronic Community Health Information System (eCHIS) [7,8] and District Health Information System (DHIS-2) [9–12] are used at Health Post and facility levels respectively. However, the software that is currently being used in Ethiopia, including eCHIS and DHIS, has a reporting component, but these systems are lacking in features like real-time interactive visualization and a data-driven engine for evidence-based insights. As a result, it was challenging to follow and monitor the patient flow in real time. The provision of services based on data-driven evidence using current records is not practical.

Patient flow is defined as the movement of patients through a healthcare facility from the point of admission to the point of discharge [13–16]. Patient flow is broadly categorized into outpatient and inpatient flow [17–19]. An outpatient patient flow is a person who receives medical care on an outpatient basis, i.e. without spending the night in a hospital or other inpatient facility. Regardless of the length of stay, a patient who has been admitted to the hospital for an overnight stay is referred to as an inpatient. Patients in Ethiopia go from primary healthcare facilities to primary hospitals, general hospitals, and then tertiary hospitals [20]. Two elements influence patient flow: self-referrals for effective and efficient services and referrals from a health facility for the best possible care. However, inefficient patient visits and the inability to efficiently collect, visualize, and interpret patient flow data were the challenges of the South Western Ethiopian healthcare network cluster. Additionally, a medical center's waiting room is backed up due to an imbalance between personnel numbers and service demand.

On the one hand, automated systems have been employed to track patients, allocate tasks and beds, estimate and distribute budgets, regulate patient and data management, as well as monitor referral flow, to address the challenges [21–27]. On the other hand, recent developments in artificially enabled data-driven applications and visualization offer new perspectives

on how to analyze and use patient flow effectively. To improve patient flow during infectious disease outbreaks, Bishop, J. A. et al. 2021 [28] used machine learning to identify and rank the readiness of individual patients for discharge in real-time. For instance, Hall. R. 2013 [29] makes an effort to reduce healthcare delivery delays. In all, a quality improvement tool is used to help identify patient flow inefficiencies at any type of healthcare facility and inform areas for intervention to help improve care delivery processes [30–33].

However, data-driven patient flow monitoring and tracking systems are still not widely implemented in low-resource settings. Lack of adequate infrastructure, data readiness, and automated patient flow management systems are some of the challenges. Therefore, it is crucial to analyze, predict, and visualize patient flow properly to manage the frequent surge plans, prevent overcrowding in the emergency room, reduce waiting time and delay, improve staff schedules to match demand, increase the number of patients admitted to the appropriate inpatient unit based on a patient's clinical condition, and use case management. This study aims to visualize and predict the patient flow in a South Western Ethiopia healthcare network cluster using aggregated and historical data as the initial stage in designing and creating a real-time patient flow data-driven analytical dashboard in Ethiopia. Furthermore, the current digital dataset is only available in aggregated data format, while the raw datasets are still in paper-based format. This study also employed the random forest [34,35] and K-nearest neighbors [36,37] algorithms to model and predict outpatient flow based on historical and aggregated records.

## 2. Methods

The patient flow analysis and visualization were performed in a healthcare network cluster in South Western Ethiopia using historical and aggregated data.

### 2.1. Ethical considerations

We obtained ethical clearance from the Institutional Review Board (IRB) of the Institute of Health at Jimma University by the principles of research ethics. The ethical clearance was received as part of a wider research program within the scope of an all-encompassing study aimed at advancing clinical decision support for point-of-care scenarios in resource-poor settings, with special attention given to patient flow modeling and visualization. The reference number assigned to us by the IRB is IHRPGI/467/2019. The study utilized aggregated data collected from 21 hospitals and health centers. The data does not contain any individual patient identifiers, ensuring the privacy and confidentiality of the study. Furthermore, since the study solely relied on historical aggregated data, explicit consent from participants was not deemed necessary.

### 2.2. Data sources

The Ethiopian health system is now employing DHIS for administrative, disease, and service data (including outpatient and inpatient flow disease). The data source for the study is the District Health Information System (DHIS-2). The patient flow aggregated data includes 41 months, from the Gregorian calendar's 2019 to 2022 (G.C.) (or the Ethiopian calendar's September 2011 to January 2014 E.C.).

### 2.3. Data collection and sampling procedures

The patient flow datasets were collected from the South Western Ethiopian healthcare network cluster. All deliveries from health centers (HC), primary hospitals (PH), general hospitals (GH), and specialized hospitals (SH) have been collected since DHIS2 was implemented. The two deliveries and reports, such as the inpatient and outpatient flows, were the only ones we

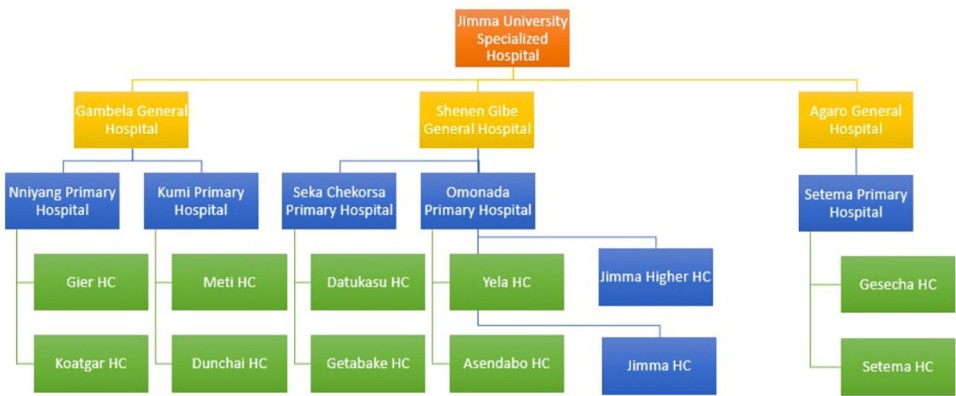

**Fig 1. Patient flow data and collection health centers and hospitals.**

focused on for this study. The selection of HC, PH, GH, and SH as well as sampling were purposefully chosen. Based on the patient flow, this study comprised a total of 21 hospitals and health centers from the South Western Ethiopian healthcare network. Fig 1 depicts the details of the hospitals and health centers as well as the hierarchy of patient flow from health centers to specialized hospitals. Moreover, patients routinely move between health centers and primary hospitals, primary hospitals and general hospitals, and general hospitals and specialized hospitals. In all, this study included a total of twelve HCs, five PHs, three GHs, and one SH. The details of the hospitals and health centers are as follows:

- Jimma University Medical Center (Specialized Hospital),

- Three General Hospitals: Gambella General Hospital, Agaro General Hospital, and

- Five Primary Hospitals: Nyinenyang Primary Hospital, Kumi Primary Hospital, Seka Chekorsa Primary Hospital, OmoNada Primary Hospital, Setema Primary Hospital,

- Twelve health centers

However, this study only considers and assumes a direct patient flow or movement, as depicted in Fig 1, while maintaining the hierarchy from a health center to a primary hospital, a general hospital, and lastly a specialty hospital. We neither implicitly nor explicitly take into account a direct patient movement from health centers to general or specialty hospitals.

## 2.4. Data preprocessing

The monthly aggregated patient flow data was first collected from 21 health centers, general hospitals, primary hospitals, and specialized hospitals. Age group, month, year, gender, category, source, name, referral in from, referral out to, outpatient visits, inpatient admissions, admission rate, bed occupancy rate, length of the stay reporting period in days, the total number of beds, number of inpatient discharges, and total length of stay in days during discharge are all included in the final processed data for patient flow prediction. Table 1 presents the details of each attribute and description.

## 2.5. Patient flow analysis, visualization, and modeling

Python was used for data preprocessing and analysis. Sankey diagrams [38,39] were used to create a patient flow visualization. The Python code for the patient flow visualization is

**Table 1. Attributes and their descriptions.**

| Attributes | Descriptions |
|---|---|
| Name | The name of the Health Centers, Primary hospitals, General hospitals, and Specialized hospitals. |
| Month | The specific month |
| Year | The specific year. The patient flow data covers from 2011–2014 E.C (2019–2022 G.C) |
| Age Group | The specific age group of patient flow <5, 5–10, 11–19, 20–29,30–45,46–65, and > = 66. |
| Gender | Male or Female |
| Category | Whether it's a Specialized hospital, General Hospital, Primary hospital, or Health center. |
| Source | The origin where the patient flow initiates i.e. whether it is from Health centers, Primary hospitals, General hospitals, and Specialized hospitals. |
| Referral in from | Origin of patient flow |
| Referral out to | Destination of the patient flow |
| Outpatient visits | Total number of outpatient visits |
| Inpatient admissions | Total number of inpatient admissions |
| Admission rate | Total number of admission rate |
| Bed occupancy rate | The bed occupancy rate in specific Health centers, Primary hospitals, General hospitals, or Specialized hospitals. |
| Length of stay reporting period in days | Length of stay reporting period in days in specific Health centers, Primary hospitals, General hospitals, or Specialized hospitals. |
| Total number of beds | Total number of beds in Health centers, Primary hospitals, General hospitals, or Specialized hospitals. |
| Number of inpatient discharges | The number of inpatient discharges in Health centers, Primary hospitals, General hospitals, or Specialized hospitals. |
| The total length of stay in days during discharge | The total length of stay in days during discharge in Health centers, Primary hospitals, General hospitals, or Specialized hospitals. |

available on https://github.com/gel1has3/Patient_Flow_Analysis_and_Prediction. Patient flow visualization was analyzed and visualized using the following criteria: category, name, age group, gender, and total number of flows. Overall, the total number of *"outpatient visits"*, *"inpatient visits"*, *"admission rate"*, and *"duration of stay"* were examined and depicted using the Snakey diagram. A Sankey diagram, a specific type of flow chart, displays the volume of patients moving between the health center and hospital according to the width of the paths to enable category relationships and flow visualization.

## 2.6. Modeling patient flow

A set of steps were conducted to model and predict the patient flow. First, the dataset's outpatient flow distribution was visualized using a kernel density estimate (KDE) plot [40]. Then, we used the method of Vilkko, Riitta, et al. 2021 [41] to classify the patient flow into quiet months (50% of the mean), optimal months (>50% of the mean to two times the mean), and busy months (two times the mean or more) because the distribution of the outpatient flow was not gaussian (a bell curve), and the mean score error was fairly significant.

Following that, we used a random forest approach, which is a cutting-edge state-of-the-art ensemble learning outcome in the majority of datasets, to model the aggregate dataset for patient flow. We also utilized k-nearest neighbors, which do not make any assumptions about the distribution of the aggregated patient flow and historical data. The dataset was divided into training and testing tests for the training.

The test dataset included one-third of the dataset. We used the default parameters for the random forest decision tree classifier, including the values of n_estimators, min_samples_leaf, and min_samples_split, which are equal to 100, 1, and 2, respectively. In addition, we employed hyperparameter tuning with different values, including n_estimators, max_depth, min_samples_split, and min_samples_leaf. The n_estimators of 50, 100, and 150 were used. The max_depths were None, 10, and 20. Min_samples_split was 2, 5, and 10. The min_samples_leaf was 1, 2, and 4. Whereas the hyperparameter optimization was used in KNN with n_neighbors values of 3, 5, and 7 as well as weights with a value of uniform and distance. Finally, the performance of the trained machine learning model was evaluated using accuracy and weighted F1-score. The ROC curve was used to determine the trade-off between the true positive rate (TPR) and the false positive rate (FPR) for each class. We also used the one-versus-the-rest (OVR) technique to solve the multi-class problem. OVR allows us to use existing algorithms developed for binary classification problems by dividing the classes into discrete binary classifications. The OVR allowed us to address the challenges of the multi-class situation while still capitalizing on our chosen approach's strengths.

Furthermore, an imbalance in the outpatient flow data was observed, and to address it, the NearMiss Algorithm—Undersampling [42,43], MOTE (Synthetic Minority Oversampling Technique)—Oversampling [44], and SMOTE-Tomek [45,46] techniques were applied. The performance of the results was compared to that of the original dataset.

## 3. Results

Each health center or hospital generated a total of 49960 aggregated records for 622 deliveries for 41 months. A total of 44 attributes and 1099k historical and aggregated deliveries from 21 hospitals and healthcare centers were retrieved. From these deliveries and records, there were a total of 11760 outpatient flow records with the following attributes: category, name, referral in from, referral out to, age group, gender, activity, month, year, and the number of outpatient flow visits. A total of 3528 outpatient flows were recorded in 2020 and 2021. In 2019 and 2022, respectively, there were 3234 and 1470 outpatient flows retrieved. The findings showed that there were almost equal numbers of outpatient flow interims for males and females. The number of outpatient referrals received by health posts, health centers, primary hospitals, and general hospitals was 6720, 2800, 1680, 1120, and 560, respectively. On the other side, primary hospitals, general hospitals, and specialized hospitals referred out 5600, 3920, and 2240 patients respectively.

Whereas, the number of inpatient records containing attributes including category, name, visit type, activities, and total inpatient flow was just 147. The average length of stay, the total length of stay (in days) during discharge, the total length of stay (in days) during the reporting period, the total number of beds, bed occupancy rate, total number of inpatient discharges, and admission rate were the types of inpatient activities that were collected.

### 3.1. Patient flow visualization

The patient flow visualization was categorized into outpatient flow, inpatient flow, admission rate, length of stay in days during discharge, and bed occupancy rate.

**3.1.1. Outpatient flow.** The highest volume of outpatient visits occurred in the health posts and health centers. The most outpatient patient flow was seen between Setema Primary Health Hospital and Agaro General Hospital. The outpatient flow visualization from 2019 to 2022 in the healthcare network cluster in southwest Ethiopia is presented in Fig 2.

**3.1.2. Inpatient flow.** Between general hospitals and specialized hospitals as well as between general hospitals and primary health hospitals, the most inpatient flow was seen. The

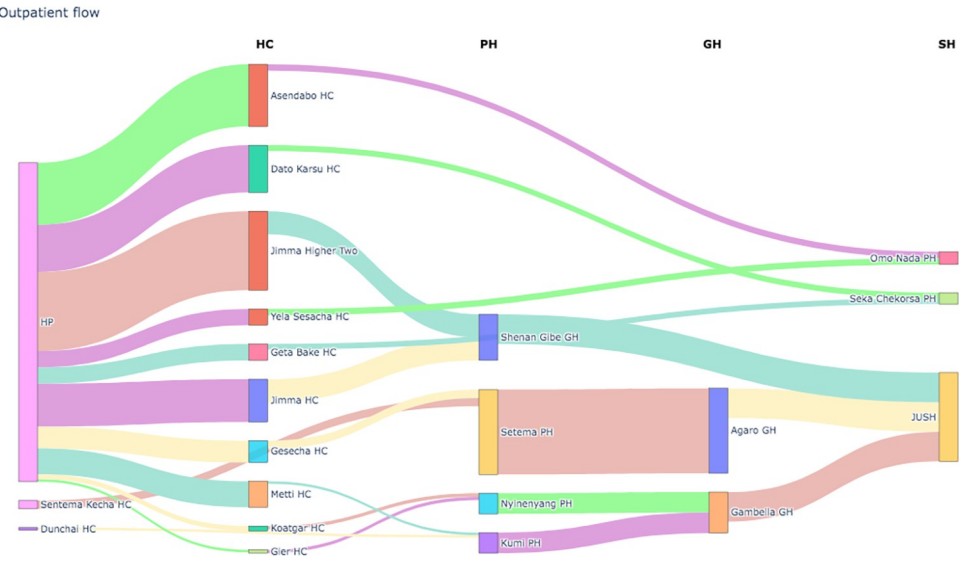

**Fig 2. Outpatient flow visualization from 2019 to 2022.**

highest inpatient flow, for instance, was seen between Setema PPrimary Hospitals and Agaro general hospitals. Following it, Shenan Gibe Hospitals, Agaro General Hospital, and Gebmbella General Hospital have all sent a sizable number of inpatients to Jimma Specialized Hospital. The inpatient flow visualization from 2019 to 2022 in the healthcare network cluster in southwest Ethiopia is presented in Fig 3.

**3.1.3. Inpatient admission.** Between primary hospitals and general hospitals, as well as general hospitals and specialized hospitals, inpatient admissions were found to be the highest. However, the health center had the lowest inpatient admissions. The highest inpatient admission was observed from Shenan Gibe Hospital to Jimma Specialized Hospitals, with 23.55%. Fig 4 depicts the inpatient admissions in the healthcare network cluster in southwest Ethiopia from 2019 to 2022.

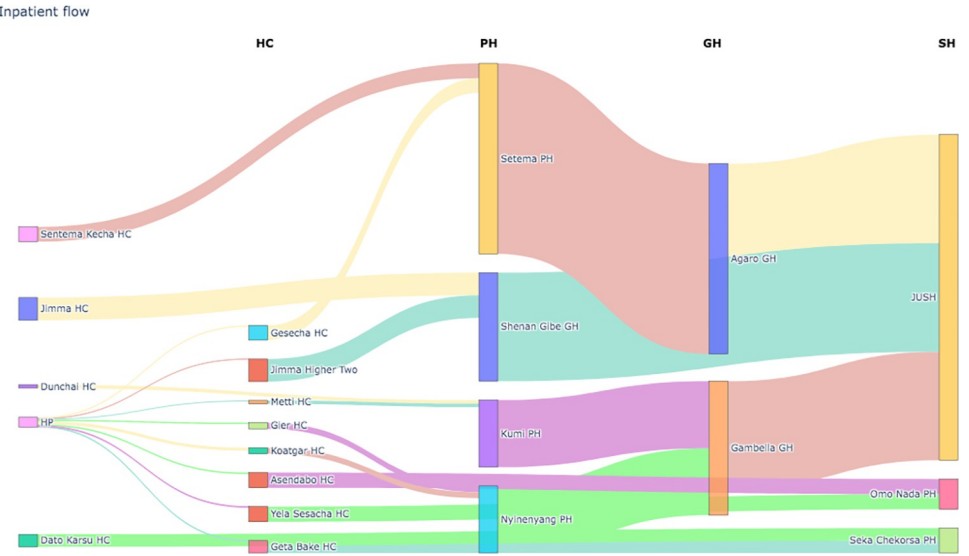

**Fig 3. Inpatient flow visualization from 2019 to 2022.**

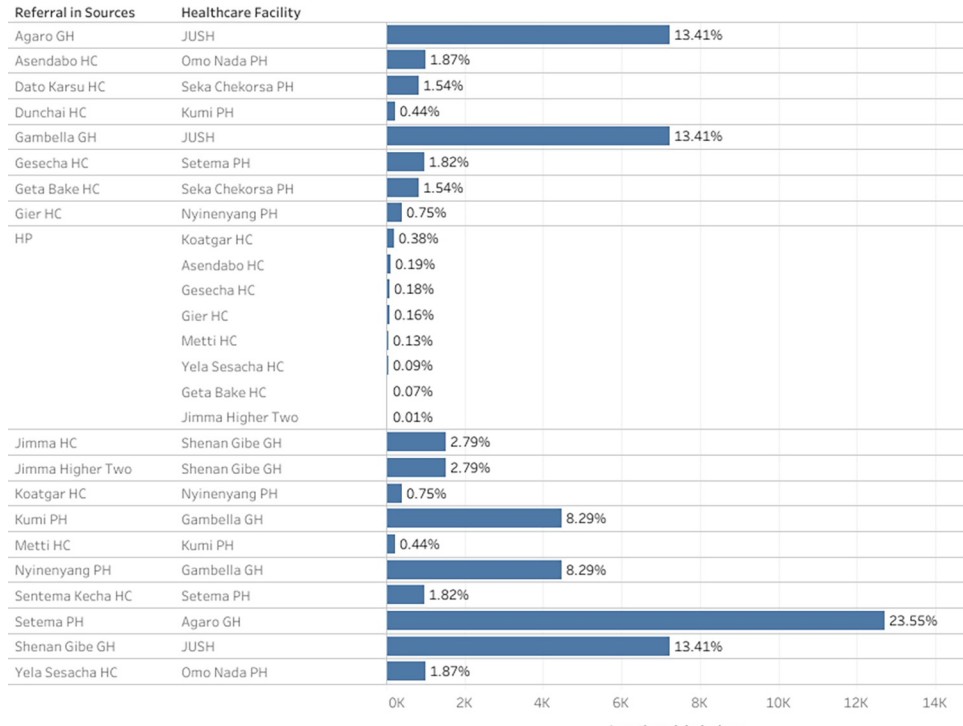

**Fig 4. Inpatient admissions in the healthcare network cluster in southwest Ethiopia from 2019 to 2022.**

**3.1.4. Bed occupancy rate and length of stay in days during discharge.** The highest bed occupancy rate and length of stay in days during discharge were observed in specialized and general hospitals. A minimum length of stay was found in the health centers. Setema Primary Hospital and Agaro General Hospitals recorded the longest stays. The length of stay was highest in general and specialized hospitals such as Agaro, Gambella, Setema, and Shannan Gibe General Hospitals, as well as Jimma University Specialized Hospital.

Whereas, the primary hospital to general hospital has the greatest bed occupancy rate. We found that Setema Primary Hospital and Agarao General Hospitals had the highest bed occupancy admission rates. Furthermore, we also observed that the Sentema Kecha Health Center and the Setema main hospitals had the greatest bed occupancy rates. Jimma's Higher Two and Jimma's Health Center had the lowest bed occupancy rate.

## 3.2. Patient flow prediction

**3.2.1. Experimental r esults.** The patient flow dataset had an outpatient flow mean of 235. The outpatient flow was not normally distributed. Then, based on the computation of the outpatient flow mean, we categorized the number of outpatient flows into quiet, optimal, and busy monthly outpatient flows to describe the unevenness of the patient flow. The cutoff for a busy monthly outpatient flow was greater than 470, whereas the range for an optimal monthly outpatient flow was 235 to 470. The threshold for a quiet monthly outpatient flow was 235. In all, there were respectively 8590, 1682, and 1488 quiet, optimal, and busy outpatient flows observed. More details on the visual representation of the outpatient flow are provided in Fig 2. The k-nearest neighbors algorithm had an accuracy of 76.41% compared to the decision tree's 78.43%. The detailed experimental results of the forecasting and modeling of outpatient flow are shown in Tables 2 and 3.

**Table 2. Presents the detailed results of the experimentation.**

| Classifier | Strategy | Accuracy | Weighted F1-score |
|---|---|---|---|
| **Random Forest** | Hyper-parameter Tuning | 0.868 | 0.868 |
| | OneVsRestClassifier(OVR) | 0.868 | 0.868 |
| KNN | Hyper-parameter Tuning | 0.840 | 0.839 |
| | OneVsRestClassifier(OVR) | 0.840 | 0.839 |

**Table 3. Presents the detailed results of the experimentation using data imbalance techniques.**

| Category | Status | Distribution outpatient flow | Algorithm | Accuracy |
|---|---|---|---|---|
| Oversampled dataset | Balanced with SMOTE Oversampling | quiet = 8590 optimal = 8590 busy = 8590 | Random Forest Decision Tree | 0.91 |
| | | | KNN | 0.86 |
| Under sampled dataset | Balanced with NearMiss Algorithm–Undersampling | quiet = 1488 optimal = 1488 busy = 1488 | Random Forest Decision Tree | 0.69 |
| | | | KNN | 0.64 |
| Dataset based on over- and under-sampling using SMOTE and Tomek links | Balanced with SMOTETomek | quiet = 8554 optimal = 8543 busy = 8543 | Random Forest Decision Tree | 0.91 |
| | | | KNN | 0.86 |

The model achieved an accuracy of 0.868 and a weighted F1 score of 0.868. These metrics indicate the overall predictive capability and balance between precision and recall. The ROC curve is presented in Figs 5 and 6, illustrating the trade-off between true positive rate (TPR)

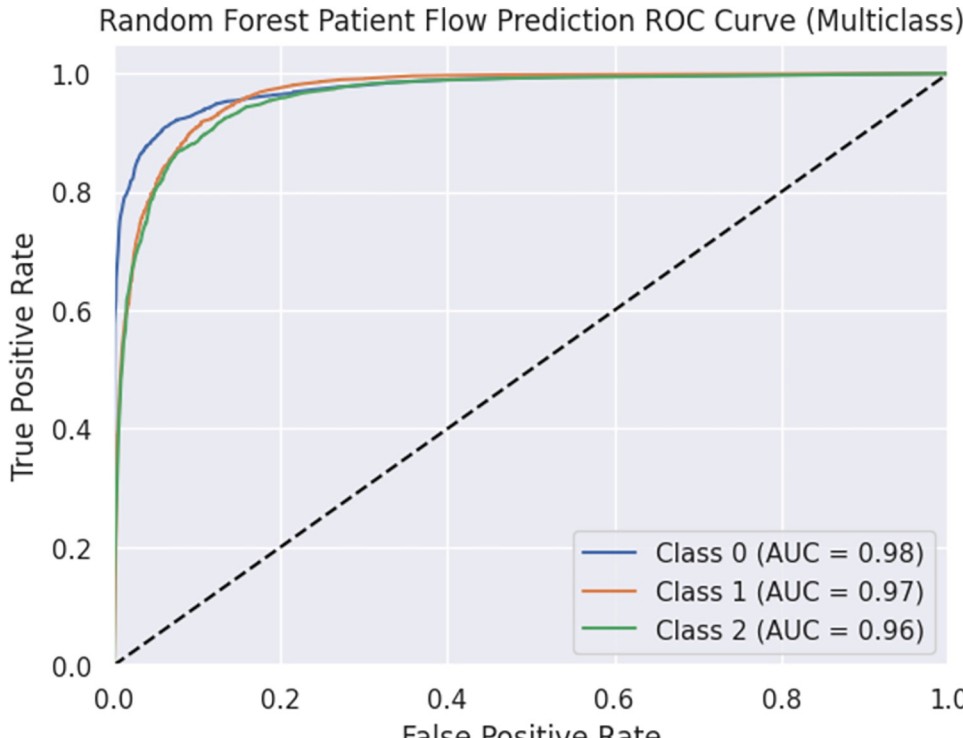

**Fig 5. Random Forest Patient Flow Prediction ROC Curve (Multiclass).**

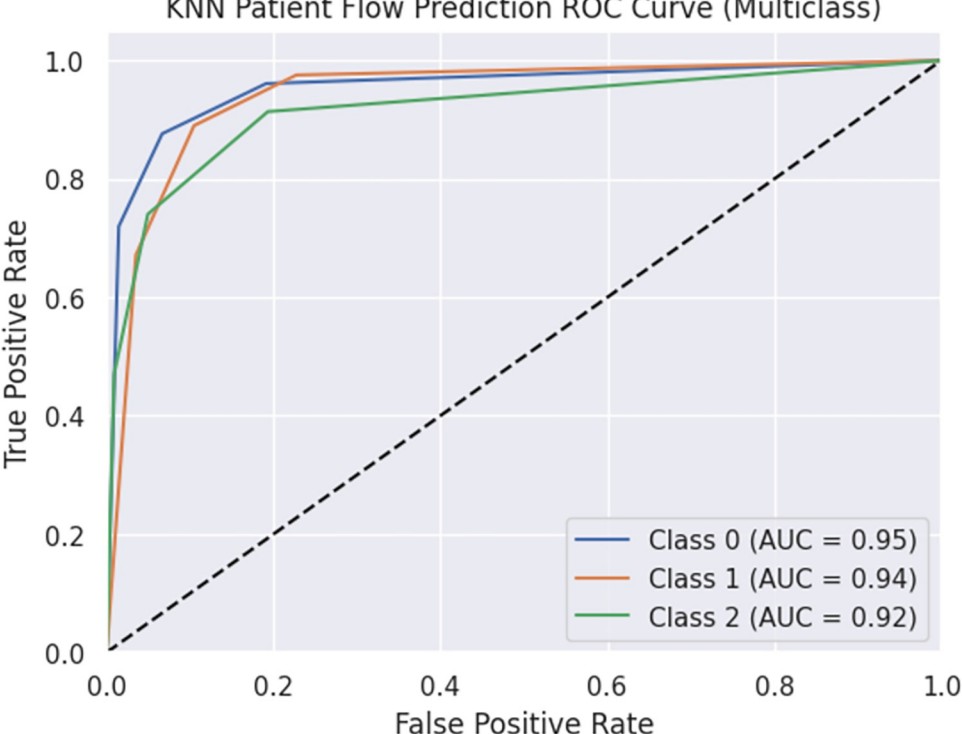

**Fig 6. KNN Patient Flow Prediction ROC Curve (Multiclass).**

and false positive rate (FPR) for each class. The area under the curve (AUC) values for Classes 0, 1, and 2 were greater than 0.90. The ROC curve showcases the model's ability to distinguish between different classes and detailed information is found in Figs 5 and 6. Furthermore, the OVR ROC curve is depicted in Figs 7 and 8.

## 4. Discussion

The study was set up to visualize and predict the patient flow in a South Western Ethiopia healthcare network cluster. Furthermore, the study employed the random forest and K-nearest neighbors algorithms to model and predict outpatient flow based on historical and aggregated records. Records of patient flow or deliveries for a total of 41 months were collected from 21 hospitals and healthcare facilities. Following the present results, there were respectively 8590, 1682, and 1488 quiet, optimal, and busy outpatient flows observed. The accuracy of the k-nearest neighbors method was 76.41%, whereas the accuracy of the decision tree was 78.43%.

Health posts and health centers saw the highest amount of outpatient visits. The data-driven patient flow visualization and prediction tool may be recommended to assist with continuous surge planning, reducing emergency department overcrowding, lowering long wait times, and maximizing staffing levels to handle the demand. This will help the hospital and health centers achieve the principle of "*the hospital and health centers starts with zero outpatient and ends with zero outpatient*" [33].

The highest inpatient flow was seen between general hospitals and primary hospitals, as well as between general hospitals and specialist hospitals. In the health centers, a minimal stay period was observed. Hospitals with general and specialized services had the longest stays. The highest admission rates were seen between primary hospitals and general hospitals as well as

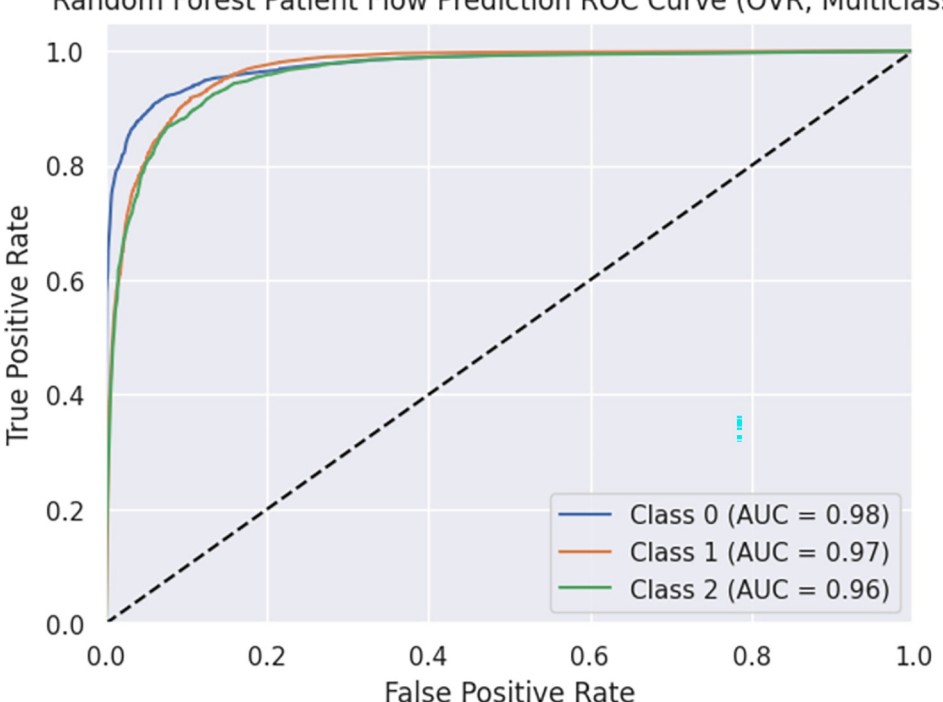

**Fig 7. Random Forest Patient Flow Prediction ROC Curve (OVR, Multiclass).**

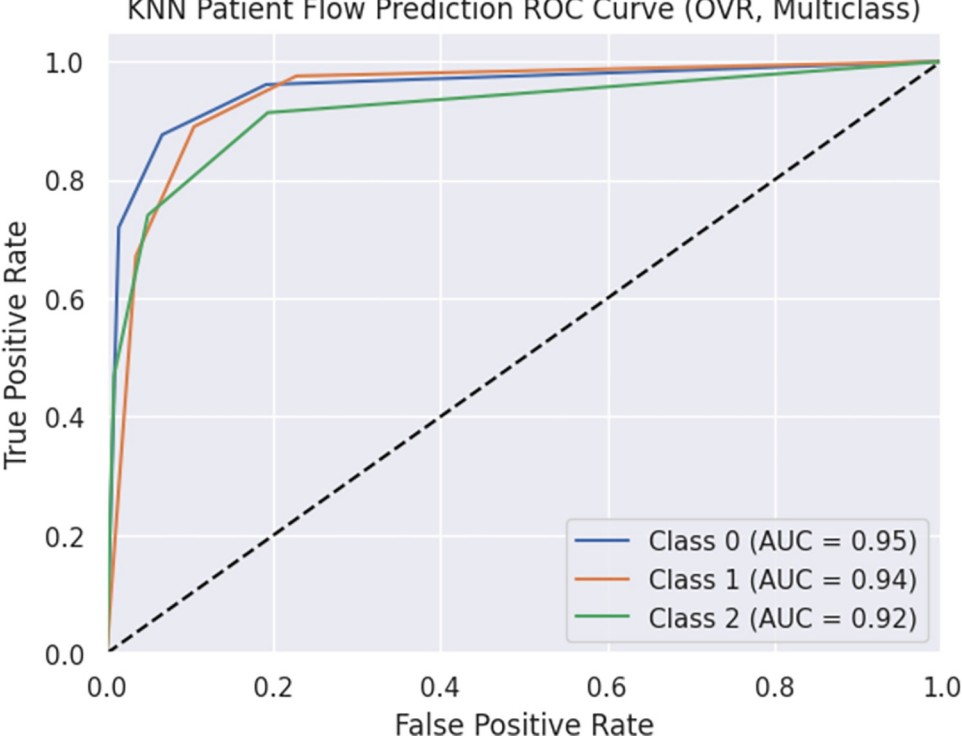

**Fig 8. KNN Patient Flow Prediction ROC Curve (OVR, Multiclass).**

general hospitals and specialized hospitals. The primary hospital to general hospital has the greatest bed occupancy rate. However, we were unable to develop an inpatient prediction model in the current study due to the small number of inpatient data. Therefore, further investigation and modeling are required to predict inpatient flow, including the average length of stay, the total length of stay (in days) during discharge, the total length of stay (in days) during the reporting period, the total number of beds, the bed occupancy rate, the total number of inpatient discharges, and the admission rate.

Overall, our findings prompt interesting considerations about the type and scope of patient flow visualization and prediction in a Western Ethiopian healthcare network cluster. Unfortunately, we were only able to visualize inpatient and outpatient flows. Additionally, a prediction model for outpatient flow based on random forest and KNN algorithms was developed.

Our study has limitations. To collect the patient flow records from the DHIS2 historical reports and deliveries, we purposefully sampled 21 hospitals and health centers. We did not, however, examine the DHIS2-aggregated data's quality. Moreover, this study is only limited to monthly aggregated patient flow data. This study solely takes into account and assumes a direct patient flow or movement while preserving the hierarchy from a health center to a primary hospital, a general hospital, and finally a specialty hospital.

## 5. Conclusion

In this study, patient flow in a South Western Ethiopian healthcare network cluster was visualized and predicted using historical and aggregated data collected over 41 months. A total of 21 health centers and hospitals were utilized to study and predict the patient flow of the healthcare network cluster in southwest Ethiopia. Outpatient flow and inpatient flow, as well as admission rate, length of stay in days during discharge, and bed occupancy rate, were included in the patient flow visualization categories. The patient flow prediction, however, was only conducted for outpatient flow cases since the inpatient flow data were so sparse. We achieved an accuracy of 0.85 and 0.83 for the outpatient flow modeling and prediction using the random forest and KNN algorithms, respectively.

The study made several meaningful observations: (1) It is feasible to apply data-driven solutions to patient flow prediction; (2) The need for and amount of data created by these digital tools will grow along with their use, demanding effective data-driven visualization and prediction to support evidence-based decision-making; and (3) Modeling patient flow is essential for minimizing the need for ongoing surge plans, preventing emergency department overcrowding, eliminating lengthy wait times, and optimizing staffing levels to handle the demand. In conclusion, we developed a patient flow visualization and prediction model as a first step toward an end-to-end effective real-time patient flow data-driven and analytical dashboard in Ethiopia, as well as a plugin for the already-existing digital health information system. Moreover, future research might also consider the cost-benefit trade-offs of data visualization at scale in low-resource settings because the existing digital tool will generate a lot of data in the future. The severity of cases, the length of consultations, and the distance from the hospital and healthcare facilities, among other factors, must be taken into consideration in addition to classifying the patient flow based on busy, optimal, and quiet scenarios.

## Acknowledgments

The Capacity Building and Mentorship Program (CBMP), Jimma University, Jimma, Ethiopia is gratefully acknowledged by the authors for providing data access and the necessary logistics. We would like to express our deepest gratitude to Mr. Asaye for his remarkable cooperation in facilitating data access and collection process.

## Author Contributions

**Conceptualization:** Balew Ayalew Kassie, Geletaw Sahle Tegenaw.

**Data curation:** Balew Ayalew Kassie, Geletaw Sahle Tegenaw.

**Formal analysis:** Balew Ayalew Kassie, Geletaw Sahle Tegenaw.

**Funding acquisition:** Balew Ayalew Kassie, Geletaw Sahle Tegenaw.

**Investigation:** Balew Ayalew Kassie, Geletaw Sahle Tegenaw.

**Methodology:** Balew Ayalew Kassie, Geletaw Sahle Tegenaw.

**Project administration:** Balew Ayalew Kassie, Geletaw Sahle Tegenaw.

**Resources:** Balew Ayalew Kassie, Geletaw Sahle Tegenaw.

**Software:** Balew Ayalew Kassie, Geletaw Sahle Tegenaw.

**Supervision:** Balew Ayalew Kassie, Geletaw Sahle Tegenaw.

**Validation:** Balew Ayalew Kassie, Geletaw Sahle Tegenaw.

**Visualization:** Balew Ayalew Kassie, Geletaw Sahle Tegenaw.

**Writing – original draft:** Balew Ayalew Kassie, Geletaw Sahle Tegenaw.

**Writing – review & editing:** Balew Ayalew Kassie, Geletaw Sahle Tegenaw.

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
