## [Decision Letter · Decision Letter 0]

11 Jul 2023

PDIG-D-23-00182

Developing a Patient Flow Visualization and Prediction Model using Aggregated Data for a Healthcare Network Cluster in Southwest Ethiopia

PLOS Digital Health

Dear Dr. Kassie,

Thank you for submitting your manuscript to PLOS Digital Health. After careful consideration, we feel that it has merit but does not fully meet PLOS Digital Health's publication criteria as it currently stands. Therefore, we invite you to submit a revised version of the manuscript that addresses the points raised during the review process.

Please submit your revised manuscript within 60 days Sep 09 2023 11:59PM. If you will need more time than this to complete your revisions, please reply to this message or contact the journal office at digitalhealth@plos.org. Please include the following items when submitting your revised manuscript:

We look forward to receiving your revised manuscript.

Kind regards,

Young-Gab Kim, Ph.D.

Academic Editor

PLOS Digital Health

Journal Requirements:

Additional Editor Comments (if provided):

Based on the reviewers' comments after a careful review process and an independent in-depth reading of your manuscript, we recommend a MAJOR REVISION.

When revising your manuscript, please consider all issues mentioned in the reviewers' comments carefully: please outline every change made in response to their comments and provide suitable rebuttals for any comments not addressed.

Reviewers' comments:

Reviewer's Responses to Questions

**Comments to the Author**

1. Does this manuscript meet PLOS Digital Health’s publication criteria? Is the manuscript technically sound, and do the data support the conclusions? The manuscript must describe methodologically and ethically rigorous research with conclusions that are appropriately drawn based on the data presented.

Reviewer #1: No

Reviewer #2: Partly

2. Has the statistical analysis been performed appropriately and rigorously?

Reviewer #1: No

Reviewer #2: No

3. Have the authors made all data underlying the findings in their manuscript fully available (please refer to the Data Availability Statement at the start of the manuscript PDF file)?

Reviewer #1: Yes

Reviewer #2: Yes

4. Is the manuscript presented in an intelligible fashion and written in standard English?

Reviewer #1: Yes

Reviewer #2: Yes

5. Review Comments to the Author

Reviewer #1: Patient flow in healthcare systems demonstrates patient interaction or movement through various service points (from the point of registration to discharge common for both inpatients and outpatients). Therefore effective health information systems depict various patient interaction points. Identifying challenges at each of these interaction points is a sure way of dealing with patient flow problems such as overcrowding, scheduling, reducing queueing/waiting time, improving quality of service etc at a specific service point. (World Health Organization, “Strengthening health information systems,” Strength. Heal. Inf. Syst., pp. 1–4, 2017) 

(Q. Nguyen, M. Wybrow, F. Burstein, D. Taylor, and J. Enticott, “Understanding the impacts of health information systems on patient flow management: A systematic review across several decades of research,” PLoS One, vol. 17, no. 9 September, pp. 1–20, 2022, doi: 10.1371/journal.pone.0274493.)

Author description of patient flow both in text (abstract section), results and figure 1 in this context can be described as statistical description of hospital attendance and are inconsistent with patient interactions (flow) in any health information system.

Reviewer #2: In this research article, the authors implement two machine learning methods to predict the outpatient flow from a number of health facilities in Ethiopia, and use Sankey diagrams to visualize these flows. The article is clear and easy to follow, and the problem it seeks to address is both important and well characterized. However, I have two major comments:

1. The statistical modeling component of this paper would benefit from additional description, for example:

a. Details about testing and training datasets (e.g. how were training/test sets divided, was cross-validation employed?)

b. Model hyperparameters (e.g. how many trees were in the RF, how many neighbors were used in the KNN model, how were hyperparameters optimized?)

c. How were the thresholds between quiet, optimal, and busy determined, and what is the utility of binning the flows this way instead of using a continuous or binary outcome?

d. Besides accuracy, how did these models perform? Accuracy is not an appropriate measure of model performance on its own - recommend also reporting AUROC, FPR, TPR, etc. Also consider using a one-versus-rest (OVR) approach to this multiclass problem

e. I recommend not balancing the data (see https://doi.org/10.1093/jamia/ocac093 for example), especially if there is interest in interpreting these models. Instead, it is better to choose a different decision threshold and/or use a different performance measure than accuracy (see point d)

2. Regarding the visualization component of this article, it is not clear what the different Sankey diagrams are supposed to demonstrate. Sankey plots are primarily for visualizing the flow of some quantity, but here they are used also to show admission rate, length of stay, and bed occupancy rate. These quantities may be better visualized using other plot types. I also suggest further discussion of the diagrams in the body of the text.

6. PLOS authors have the option to publish the peer review history of their article (what does this mean?). If published, this will include your full peer review and any attached files.

**Do you want your identity to be public for this peer review?** For information about this choice, including consent withdrawal, please see our Privacy Policy.

Reviewer #1: Yes: Michael Owusu-Adjei

Reviewer #2: No

---

## [Decision Letter · Decision Letter 1]

23 Sep 2023

Developing a Patient Flow Visualization and Prediction Model using Aggregated Data for a Healthcare Network Cluster in Southwest Ethiopia

PDIG-D-23-00182R1

Dear Mr. Kassie,

We are pleased to inform you that your manuscript 'Developing a Patient Flow Visualization and Prediction Model using Aggregated Data for a Healthcare Network Cluster in Southwest Ethiopia' has been provisionally accepted for publication in PLOS Digital Health.

Best regards,

Young-Gab Kim, Ph.D.

Academic Editor

PLOS Digital Health

Reviewer Comments (if any, and for reference):

Reviewer's Responses to Questions

**Comments to the Author**

1. If the authors have adequately addressed your comments raised in a previous round of review and you feel that this manuscript is now acceptable for publication, you may indicate that here to bypass the “Comments to the Author” section, enter your conflict of interest statement in the “Confidential to Editor” section, and submit your "Accept" recommendation.

Reviewer #1: All comments have been addressed

Reviewer #2: All comments have been addressed

2. Does this manuscript meet PLOS Digital Health’s publication criteria? Is the manuscript technically sound, and do the data support the conclusions? The manuscript must describe methodologically and ethically rigorous research with conclusions that are appropriately drawn based on the data presented.

Reviewer #1: Yes

Reviewer #2: Yes

3. Has the statistical analysis been performed appropriately and rigorously?

Reviewer #1: Yes

Reviewer #2: Yes

4. Have the authors made all data underlying the findings in their manuscript fully available (please refer to the Data Availability Statement at the start of the manuscript PDF file)?

Reviewer #1: Yes

Reviewer #2: Yes

5. Is the manuscript presented in an intelligible fashion and written in standard English?

Reviewer #1: Yes

Reviewer #2: Yes

6. Review Comments to the Author

Reviewer #1: Reasons given with detailed explanation for research focus is considered valid and fully addresses the problem under consideration. It therefore address concerns raised in the first review.

Reviewer #2: n/a

7. PLOS authors have the option to publish the peer review history of their article (what does this mean?). If published, this will include your full peer review and any attached files.

**Do you want your identity to be public for this peer review?** For information about this choice, including consent withdrawal, please see our Privacy Policy.

Reviewer #1: No

Reviewer #2: No
